# The Effect of Concurrent Resistance Training on Upper Body Strength, Sprint Swimming Performance and Kinematics in Competitive Adolescent Swimmers. A Randomized Controlled Trial

**DOI:** 10.3390/ijerph181910261

**Published:** 2021-09-29

**Authors:** Sofiene Amara, Tiago M. Barbosa, Yassine Negra, Raouf Hammami, Riadh Khalifa, Sabri Gaied Chortane

**Affiliations:** 1Higher Institute of Sport and Physical Education of Ksar-Said, University of La Manouba, Tunis 2010, Tunisia; Coachsofieneamara@gmail.com (S.A.); raouf.cnmss@gmail.com (R.H.); riadhkhal@yahoo.fr (R.K.); 2Research Unit (UR17JS01) Sports Performance, Health & Society, Higher Institute of Sport and Physical Education of Ksar Saîd, Universite de la Manouba, Tunis 2010, Tunisia; 3Research Center in Sport, Health and Human Development, 5000-801 Vila Real, Portugal; barbosa@ipb.pt; 4Instituto Politécnico de Bragança, Department of Sports Sciences, Campus Sta., 5301-856 Bragança, Portugal; 5Research Laboratory: Education, Motor Skills, Sports and Health (LR19JS01), Higher Institute of Sport and Physical Education of Sfax, University of Sfax, Sfax 3029, Tunisia; 6Laboratory of Cardio-Circulatory, Respiratory, Metabolic and Hormonal Adaptations to Muscular Exercise, Faculty of Medicine Ibn El Jazzar, Sousse 4002, Tunisia; sabrigaied1@gmail.com

**Keywords:** bench press, water parachute, hand paddles, velocity, stroke rate

## Abstract

This study aimed to examine the effect of 9 weeks of concurrent resistance training (CRT) between resistance on dry land (bench press (BP) and medicine ball throw) and resistance in water (water parachute and hand paddles) on muscle strength, sprint swimming performance and kinematic variables compared by the usual training (standard in-water training). Twenty-two male competitive swimmers participated in this study and were randomly allocated to two groups. The CRT group (CRTG, age = 16.5 ± 0.30 years) performed a CRT program, and the control group (CG, age = 16.1 ± 0.32 years) completed their usual training. The independent variables were measured pre- and post-intervention. The findings showed that the one-repetition maximum bench press (1RM BP) was improved only after a CRT program (d = 2.18; +12.11 ± 1.79%). Moreover, all sprint swimming performances were optimized in the CRT group (d = 1.3 to 2.61; −4.22 ± 0.18% to −7.13 ± 0.23%). In addition, the findings revealed an increase in velocity and stroke rate (d = 1.67, d = 2.24; 9.36 ± 2.55%, 13.51 ± 4.22%, respectively) after the CRT program. The CRT program improved the muscle strength, which, in turn, improved the stroke rate, with no change in the stroke length. Then, the improved stroke rate increased the swimming velocity. Ultimately, a faster velocity leads to better swim performances.

## 1. Introduction

Plenty of Olympic time-based athletes engage in strength and conditioning programs. A meta-analysis reported that strength training had a moderate positive effect on Olympic time-based sports performances (d = 0.59) [1].

In addition, several scientific reviews and position articles [2,3] revealed that resistance training represents a safe and feasible means in healthy adolescents to enhance their muscular strength and motor skills and to prevent sport injuries. More specifically, it has been reported that well-developed levels of muscle strength and power play an important role in achieving a high sprint swimming performance in adolescent and youth swimmers [2,3]. In the case of competitive swimming, optimizing the performances in short-distance events depends on a higher level of maximum upper body strength [4,5]. Several training methods have been developed to improve muscle strength in swimmers [2,6,7]. Indeed, dry land resistance training has been noted as very important for improving the maximum upper body strength in competitive swimmers [7,8]. Likewise, in-water resistance training has also been reported in several other studies [6,9].

The bench press (BP) exercise has been used extensively in dry land strength training protocols in swimmers [10,11]. Keiner et al. [5] noted that the one-repetition maximum in the bench press (1RM BP) and squat explained 50–60% of the 50-m and 100-m performances in the front crawl. In addition, training with the bench press exercise improves the strength in the majority of upper body muscles, such as the triceps brachii, pectoralis major and anterior deltoid [12]. Similarly, the arm muscles (biceps brachii and triceps brachii) and shoulder muscles (trapezius and deltoid) are among the active muscles during the front crawl swim [13]. This activation of the upper body muscles during the bench press exercise and the front crawl can be a positive factor that promotes the gaining of strength to optimize the sprint swimming performance. Training with medicine ball throwing (MBT) has also been one of the most effective exercises for improving strength and power in athletes [8,14]. Ignjatovic et al. [14] showed that the 1RM bench press was increased by 6.37% after 12 weeks of training with MBT in women handball players (age: 16.9 ± 1.2 years). In tandem, Raeder et al. [15] showed that 6 weeks of MBT training exercises can improve the throwing velocity by 14% and the isokinetic strength (concentric shoulder internal rotation at 180 s^−1^ by 15% in competitive female handball players (age = 20.8 ± 3.3 years)). Notwithstanding, the MBT and bench press exercises are recommended in dry land programs for swimmers to improve sprint swimming performances [7,11].

In-water resistance training makes use of a water parachute (WP) and hand paddles (HP). Several studies noted performance enhancements after in-water resistance programs [6,9]. For instance, Gourgoulis et al. [6] showed that 11 weeks of training with resistance sets with WP improved the velocity of the 50-m front crawl by 1.52%. Likewise, the choice of HP size is very important for better results after the intervention program. Crocker et al. [16] showed that hand paddles with sizes between 210 cm^2^ and 358 cm^2^ are effective at reducing the metabolic cost of transport while swimming (10.93–20%) compared to swimming without paddles. According to the literature, resistance exercises with aquatic equipment were generally performed over short-distances, such as 15 m and 25 m at the maximum intensity [6,9].

The effect of resistance training on dry land or in-water on kinematic variables (velocity (v), stroke rate (SR), stroke length (SL) and stroke index (SI)) was studied in several investigations [6,7,9,17]. Morais et al. [17] showed that 34 weeks of a training program included dry land resistance (upper and lower strength) and improved the SR (4.88%) and V (3.60%) in the 100-m front crawl in competitive swimmers (age = 13.3 ± 0.85 years). More recently, Lopes et al. [7] revealed that 8 weeks of dry land included BP and MBT and did not change the SL and SI in the 50-m front crawl (*p* > 0.05) in competitive swimmers. On the other hand, Grourgoulis et al. [6] showed that 11 weeks of resistance training in water with WP increased the velocity of the 50-m (2.18%), while the SR and SL remained unchanged (*p* > 0.05) in competitive swimmers (age = 13.08 ± 0.9 years). In the same context, Barbosa et al. [9] reported that 4 weeks of resistance training in water did not change the stroke kinematics (V, SR and SL) of the 25-m front crawl (*p* > 0.05) in competitive swimmers (age = 21.8 ± 1.9 years).

Based on the current body of literature, there is evidence that either dry land or in-water resistance training can optimize the muscle strength and, thus, enhance the sprint swimming performance. However, it is yet unclear the effect of concurrent dry land and in-water programs. As such, the present study is the first investigation to examine the effects of 9 weeks of concurrent dry land (BP and MBT exercises) and in-water (HP and WP) resistance training on the maximum upper body strength (1RM BP), sprint swimming performance (25-m and 50-m front crawl and 25-m and 50-m front crawl with arms) and kinematics in the 50-m front crawl (velocity, stroke rate, stroke length and stroke index) in competitive swimmers. It was hypothesized that 9 weeks of concurrent resistance training are sufficient to improve the maximum upper body strength and sprint swimming performance.

## 2. Materials and Methods

### 2.1. Experimental Approach to the Problem

A 9-week randomized controlled study was selected to investigate the effects of concurrent dry land and in-water resistance training on the maximum upper body strength, stroke kinematics and sprint swimming performance. Twenty-two subjects were randomly allocated into two groups (concurrent dry land and in-water resistance training group or control group that completed their usual training program). Pre- and post-tests were held to determine the effects of the concurrent program on the 1RM bench; sprint swimming performance (25-m and 50-m front crawl and 25-m and 50-m front crawl arm pulls) and stroke kinematics (velocity, stroke length, stroke rate and stroke index). Resistance strength training and testing took place at a resistance training room and a 25-m indoor pool with 27.1 °C and 25.9 °C water and air temperatures (respectively) and 64% relative humidity.

### 2.2. Participants

Twenty-two competitive male swimmers voluntarily participated in this study and were randomly allocated into two groups: the concurrent resistance training group (CRTG: *n* = 11, age = 16.5 ± 0.30 years; height: 174 ± 9.80 cm; body mass = 72.7 ± 5.30 kg) and control group (CG: *n* = 11, age = 16.1 ± 0.32 years; height: 175 ± 9.70 cm body mass 73.6 ± 5.25 kg). An a priori power analysis (G*Power 3.1.9.3, Heinrich-Heine-Universität Düsseldorf, Düsseldorf, Germany) yielded a sample size of at least 9 participants per group to detect large effects (d = 1.29), assuming a power of 0.8 and alpha of 0.05. All subjects competed at national swimming competitions for at least 6 years and had more than 6 years of in-water resistance training and 4 years of dry land resistance (Table 1).

All participants and their parents were informed of all the risks and benefits of the study and signed their written consent (legal representatives). All procedures were approved by the Unit of Research (UR17JS01) of the Higher Institute of Sports and Physical Education of Ksar Saïd, Tunisia and were in accordance with the test version of the Declaration of Helsinki.

### 2.3. Procedure

A description of the study design for both groups (pre-test, intervention period, taper period and post-test) is presented in Figure 1.

#### 2.3.1. Concurrent Resistance Training (CRT)

The CRT protocol was integrated into the training season for 9 weeks (6 weeks: intervention period and 3 weeks: taper period) in replacement of some swimming-specific drills. The program consisted of two resistance training programs, one in the water and the other on dry land. The CG completed the usual in-water program composed by 6 sessions per week of training in the water (low-intensity aerobic, high-intensity aerobic and high-intensity interval training), whereas the CRTG followed a specific training program consisting of adding a resistance series with aquatic equipment (hand paddles and water parachute) to the usual in the water program and adding two resistance training sessions per week on the upper body on dry land with the bench press (BP) and medicine ball and throw (MBT) exercises during the intervention period. In addition, all the subjects followed an introductory period for two weeks prior to the start of the experimental period, with the training containing trials with WP and HP in the water and BP and MBT on the dry land strength.

#### 2.3.2. Resistance Training in Dry Land

Two nonconsecutive sessions per week (between 60 and 75 min per session) of dry land training (Table 2) were performed by the CRTG during the intervention period [10,11]. Each session began with a 5 to 10-min standard warm-up of aerobic training (treadmill and ergometer exercises) and a specific warm-up using 10–15 repetitions of 40–60% of the 1RM bench press [18]. After the warm-up, the bench press and medicine ball throw exercises were performed with the intensity varying between 60% and 80% of the 1RM of BP and with the medicine ball weight varying between 2 and 5 kg. The number of sets varied over time from 3 to 6. The number of repetitions varied between 6 and 12 repetitions and 3 min of recovery set up between the sets and the two exercises. The volume of the external training load (ETL) was calculated by determining the volume load of the individual bench presses for each swimmer (VL BP) and the volume load of the medicine ball throws (VL MBT) according to the following equations [19]:VL BP (kg) = number of sets × number of repetitions × individual load (percentage of 1RM BP)(1)
VL MBT (kg) = number of sets × number of repetitions × load used (mass of MB)(2)

During the 3 weeks of tapering period, only one dry land session per week was held. In addition, the ETL was decreased in CRTG compared to the intervention period (−69.75% of VL BP and −68.19% of VL MBT), while the intensity of the exercise varied between 70% and 80% of the 1RM BP and between 3 and 5 kg of the MBT. The adjustment of the frequency of the sessions, the volume of the training load and the intensity of the exercises were carried out according to the recommendations found in the literature [20]. The quantification of the volume load in the intervention and taper periods is presented in Table 3.

#### 2.3.3. Resistance Training in Water

The usual training consists of 6 sessions per week with an hourly volume varying between 90 and 120 min and a distance varying between 4000 and 6000 m per session. This training protocol includes aerobic and anaerobic exercises and sprint sets. Resistance sets at the maximum speed were performed 4 times per week after the warm-up. On Monday and Thursday, the swimmers performed 3 sets × 4 reps × 15 m. On Tuesday, the swimmers performed 2 sets × 4 reps × 25 m and, on Friday, underwent 2 sets × 4 reps × 25 m. The recovery on Monday and Thursday workouts was 60 s and 5 min of rest between reps and sets, respectively, while the recovery on the Tuesday and Friday workouts was 90 s and 5 min of rest between reps and sets, respectively. Conversely, CRTG followed the same usual training program during the intervention and the taper periods, but the resistance sets were performed with aquatic equipment (Monday and Thursday with hand paddles and water parachute, Tuesday with hand paddles only and Friday with a water parachute only) [6,21]. The hand paddles with a dimension of 320 cm^2^ [9] were attached to the hand by two adjustable elastic tubes and positioned near the wrist and middle finger, while the water parachute with a dimension of 900 cm^2^ [6] was attached to the swimmer’s back to a 2-m-long rigid string with a belt that was attached around the swimmer’s waist. During the tapering period, the hourly volume, swim distance and volume of resistance sets were decreased from the intervention period by 50%, whereas the intensity remained unchanged in both groups.

#### 2.3.4. Maximum Upper Body Strength Test

The maximum upper body strength test (1RM BP) was carried out in accordance with the protocols and recommendations by others [22,23]. This protocol began with a 5-min warm-up in an aerobic regime (ergometer and treadmill), followed up by a specific warm-up consisting of a series of 8–10 repetitions at 60% of the estimated 1RM on the bench press machine. Thereafter, the swimmers had 1 min of rest and were then invited to perform a test with the estimated 1RM BP. After each valid trial, the rating perception effort RPE was determined by the swimmer [24]. In addition, 3 min of recovery was performed between the valid trials. The test was terminated in the event of failure in the execution of the movement of the exercise and/or the swimmer declared that they could not perform the new repetition, with an RPE score between 9 and 10. The 1RM BP test should not pass all 5 trials. The intraclass correlation coefficient (ICC) for the pre-test and post-test reliability was 0.80.

#### 2.3.5. Sprint Swimming Performance Tests

The swimmers underwent time trials swimming the full front crawl technique and only the front crawl arm pull. All sprint swimming performance tests were performed during the second day of the evaluation period, noted in seconds, and were determined by two expert timekeepers per stopwatch (SEIKO S120-4030, Tokyo, Japan). The best performance established by the swimmers was taken into account in the statistical analysis. Prior to the start of testing, all the swimmers completed an 800-m warm-up (600 m aerobic swimming + 200 m progressive sprint swimming). The swimmers started with the 25-m and 50-m front crawl tests, which were performed with a diving start; the starting signal was individual to each swimmer [2], and the time started as soon as the swimmer left the starting block. The intraclass correlation coefficient (ICC) for the pre-test and post-test reliability was 25-m = 0.90 and 50-m: 0.90.

For the 25-m and 50-m arm pull front crawl tests, the swimmer was asked to cover the distance at the maximum velocity and with an individual start in the water (push off the wall); to eliminate the leg movement effect, a pull buoy was placed between the legs, and we gave the order not to use leg movements. The recovery between tests was set at 5 min. The intraclass correlation coefficient (ICC) for the pre-test and post-test reliability was 25-m with arms: 0.85 and 50-m with arms: 0.82.

#### 2.3.6. Kinematic Variables

The kinematic variables were measured just during the 25-m test. To eliminate the start and end effects, only 10 m (gold standard references were positioned on both sides of the pool at the distances of 7.5 and 17.5 m) was recorded by an above-surface video camera (Sony, SNC VB 603, Tokyo, Japan; 50 Hz, full HD, 1080 p) [9]. The camera video was placed about 5 m above the water and 10 m away from the swimming lane. All video sequences were analyzed using video analysis software (Kinovea, version 0.8.15, Joan Charmant & Contrib., kinovea.org) [25]. The swim velocity (V) of each 10 m was calculated from the time taken to cover the 10 m (V = 10/t10 m). The stroke rate (SR) was computed from the time taken to complete the three consecutive stroke cycles. The stroke length (SL) was calculated as the ratio between the V and corresponding SR. The stroke index (SI) was measured by multiplying the V by the SL [26]. The intraclass correlation coefficient (ICC) for the pre-test and post-test reliability of all the kinematic variables ranged between 0.73 and 0.88.

### 2.4. Statistical Analysis

All statistical measurements were presented as the mean group (M), standard deviation (SD), mean difference, partial difference in percentage and 95% confidence interval. The baseline between-group differences were computed through independent sample *t*-tests. The normality and the sphericity of the data were verified by the Shapiro–Wilk test and Mauchly test, respectively. The intraclass correlation coefficient (ICC) was used to determine the reliability of the measurements [27]. ANOVA with repeated measures was used to determine the effect of time and time × group interaction (group: CG and CRTG × time: pre- and post-interventions). Repeated measures ANOVA (time: pre-and post-training) were used to determine the changes in the performances within the group if the group × time interaction reached the level of significance (i.e., significant F value). The effect size (ES) was determined by converting a partial eta-squared to Cohen’s d. According to Cohen [28], the ES can be classified as small (0.00 ≤ d ≤ 0.2), moderate (0.2 ≤ d ≤ 0.5) or large (d ≥ 0.5). The level of significance was established at *p* ≤ 0.05. SPSS 26.0 (SPSS Inc., Chicago, IL, USA) and was used for the statistical analyses.

## 3. Results

All the participants received treatments as allocated. Table 4 displays the mean and standard deviation for all the analyzed variables. There were no statistically significant differences in the baseline values between the two groups. In addition, no significant differences were found between both groups with regards to chronological age, height and body mass.

### 3.1. Effect of Training on Maximum Upper Body Strength

Our findings revealed that the 1RM bench press improved after 9 weeks in the CRTG (12.11 ± 1.79%; 95% confidence interval (CI) −7.79 to 3.12; *p* < 0.001; d = 2.18; large). The dry land measurements remained unchanged in the CG 212 (*p* > 0.05). The dry land measurement can be observed in Table 4 and Figure 2.

### 3.2. Effect of Training on Sprint Swimming Performance

Our results showed that the 25- and 50-m front crawl improved at the post-test only in the CRTG (−6.82 ± 0.66%, −4.22 ± 0.18%; 95% confidence interval (CI) 0.38–1.41, 0.33–1.98; *p* = 0.002, 0.009; d = 1.62, 1.3; large, respectively). Likewise, the 25- and 50-m front crawl arm pulls increased after 9 weeks in the CRTG only (−6.86 ± 0.16%; −7.13 ± 0.23%; 95% confidence interval (CI) 0.68–1.43%, 1.44–3.06%; *p* < 0.001, 0.001; d = 2.61, 2.59; large, respectively). The sprint swimming performances obtained in both groups before and after 9 weeks are presented in Table 4 and Figure 3.

### 3.3. Effect of Training on the Kinematic Variables

The statistical results revealed a significant improvement in the V10-m and SR10-m after 9 weeks of the CRTG only (9.36 ± 2.55%, 13.51 ± 4.22%; 95% confidence interval (CI) −0.26 to −0.07, −0.14 to −0.06; *p* = 0.001, *p* < 0.001; d = 1.67, 2.24; large, respectively). The SL increased in the CG only (−1.50 ± 0.65%; 95% confidence interval (CI) 0.01–0.06; *p* = 0.012; d = 1.24; large). The SI remained unchanged in both groups (*p* > 0.05). All the data of the kinematic variables are presented in Table 4 and Figure 4.

## 4. Discussion

This study examined the effect of a 9-week training program consisting of dry land training out of water with BP and MBT exercises and concurrent in-water training including resistance sets with aquatic equipment (HP and WP) on the maximum upper body strength, kinematic variables and sprint swimming performances during the front crawl. This concurrent resistance training program was compared to the usual training, which was performed by the control group. The results of the present study showed a significant increase in the maximum upper body strength, stroke kinematics (SR and V) and in all the variables related to sprint swimming performances in the CRTG after 9 weeks of concurrent resistance training, leading us to accept the hypothesis.

### 4.1. Effect of Training on Maximum Upper Strength

The results of the present study showed a large enhancement in the 1RM bench press test from the pre-test to the post-test (12.11 ± 1.79%) just after the CRT. However, no significant changes in the muscle strength were detected after regular swimming training (i.e., CG). This corresponds with the principle of training specificity, which dictates that training-related adaptations are larger when the training features (e.g., type of exercise, contraction mode and movement velocity) are aligned with the tested activity [29]. These results confirmed the findings of previous studies [7,30,31], which showed that BP training with intensities between 60% and 80% and MBT with loads between 2 and 5 kg can improve the maximum upper body strength. In the same line, Lopes et al. [7] reported that the 1RM of BP was improved by 14.92% after 8 weeks of dry land strength training using BP and MBT exercises. In addition, Garrido et al. [30] showed that an 8-week dry land program that included sets of BP with intensities between 50% and 75% of the 1RM improved the performance of the 6RM bench press (+41.67%). However, the majority of the upper body muscles (biceps brachii and triceps brachii and shoulder muscles) were activated by the bench press and medicine ball throw exercises [31]. In another context, the maximum upper body strength did not change in the CG (*p* > 0.05); the absence of dry and resistance training can explain this result observed in such a group. In summary, the inclusion of BP and MBT routines into the dry land training program does improve the maximum upper body strength of swimmers. The follow-up question that is addressed in the coming sections is if this dry land enhancement can be translated into an improvement in the in-water performance, and if so, what is the underlying mechanism?

### 4.2. Effect of Training on Sprint Swimming Performance

Based on our findings, concurrent resistance training has the potential to improve sprint swimming performance (25-m and 50-m full front crawl and 25-m and 50-m front crawl arm pull) (−4.22 ± 0.18% to −7.13 ± 0.23) in competitive swimmers. According to the previous literature, only two randomized control trials investigated the effect of resistance training with aquatic equipment on sprint swimming performances. Gourgoulis et al. [6] showed that 11 weeks of an in-water training program with resistance sets using a WP improved the 50-m, 100-m and 200-m front crawl performances (3.22–7.26%). In contrast, Barbosa et al. [9] reported that 4 weeks of training with hand paddles (3 times/week) did not affect the 50-m front crawl performance (*p* > 0.05). This discrepancy in the results may be due to several factors, such as the difference in the external training load (e.g., training volume and duration of the program) and the competitive level of the participants taking part in the aforementioned studies. Our findings revealed that both the CG and CRT improved the time trials. However, the changes were significant, with large effects in the CRT. Thus, the HP and WP may also have played a role in such improvements.

According to several published researches [5,32], the sprint swimming performance was strongly correlated (r = 0.67–0.79) with the maximum upper body strength. Keiner et al. [5] showed that the 1RM bench press and squats explained 45–62% of the variance in the sprint swimming performances (50-m and 100-m front crawl). Indeed, dry land resistance has been an effective approach when incorporated into a swimming program, because it can improve the maximum muscular strength [7,8]. Lopes et al. [7] showed that resistance training on dry land with intensities between 60% and 80% of the training load improves the upper body strength (the increase in the 1RM bench press) and 50 m in the front crawl (3.98%). The gain effect of strength in water by the transfer of dry land strength and conditioning into muscle strength propulsive in front crawl swimming after 9 weeks of concurrent resistance training may be a factor that can improve all the variables related to sprint swimming performance [33].

### 4.3. Effect of Training on Kinematic Variables

It remains to be discussed the underlying mechanisms that can explain how the improvement in strength can translate into performance enhancements. Our findings showed that training with CRT improved the velocity and stroke rate (+9.36 ± 2.55%; 13.51 ± 4.22%) but not the stroke length or stroke index. Since no investigation has studied the effect of in-water resistance training with aquatic equipment combined with dry land training on the kinematic variables, it is challenging to compare the results with other studies. However, Gourgoulis et al. [6] showed that 11 weeks of resistance training in water only with a water parachute improved the velocity by 2.18%. On the other hand, Barbosa et al. [9] revealed that 4 weeks of resistance swimming with hand paddles is insufficient to improve the swim kinematics in the 50-m front crawl (*p* > 0.05). These mixed results can be explained by the differences between the effects of the chosen aquatic equipment (HP/WP or HP + WP) and the differences in the durations of the intervention programs. In another context, dry land training can improve the kinematics [7,32]. However, Lopes et al. [7] noted that 8 weeks of resistance on dry land increased the stroke rate in the 50-m front crawl (12.20%) and the stroke index in the 100-m front crawl (5.86%) in competitive swimmers. A faster velocity in short-distance swimming is strongly correlated with the maximum upper body strength [5]. The velocity can be calculated by multiplying the stroke rate by the stroke length (V = SR × SL). Therefore, the velocity increase was due to a SR increase, as reported by others for swimmers of similar competitive levels [5,34].

This study has some limitations that warrant discussion. First, the absence of physiological and neuromuscular measurements during all the study procedures. Second, by having the CRT group do both dry land and water-based resistance training, it makes it hard to determine which exercise caused the improved strength and sprint swimming performances.

## 5. Conclusions

To sum up, a program of 9 weeks of in-water training, including resistance sets making use of aquatic equipment (WP + HP) combined with dry land resistance (BP and MBT exercises), can improve the maximum upper body strength and, therefore, the stroke rate. The increase in SR led to a faster swim and, thus, better sprint swimming performance. In addition, it can also be concluded that a dry land resistance program, including the BP and MBT concurrent to in-water resistance with WP and HD, is effective in improving the maximum strength muscle, kinematics (SR and V) and, ultimately, sprint swimming performance. Additionally, the CRT program can be effective training included by swimmers to their programs of training to improve physical and kinematic swimming.

## Figures and Tables

**Figure 1 ijerph-18-10261-f001:**
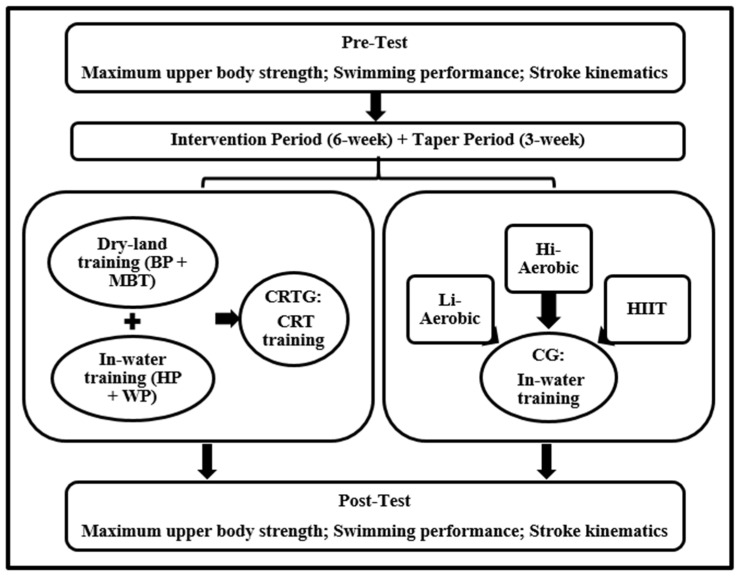
Description of the study design in both groups (CRTG: concurrent resistance training group and CG: control group). BP: bench press, MBT: medicine ball throw, Hi-Aerobic: high-intensity aerobic, Li-Aerobic: low-intensity aerobic and HIIT: high-intensity interval training.

**Figure 2 ijerph-18-10261-f002:**
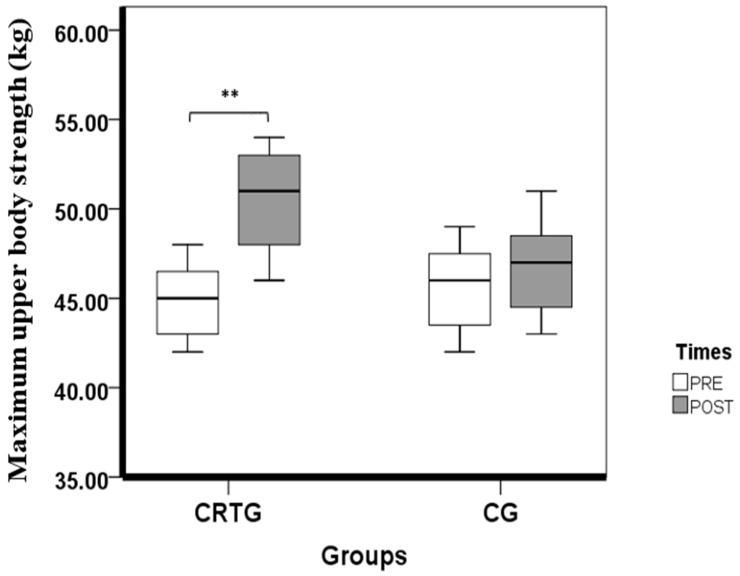
Box plots displaying the 25th and 75th percentiles, medians and whiskers extending from the minimal to maximal for the maximum upper body strength (bench press) in both groups (CRTG and CG) after 9 weeks of training. (** *p* < 0.01).

**Figure 3 ijerph-18-10261-f003:**
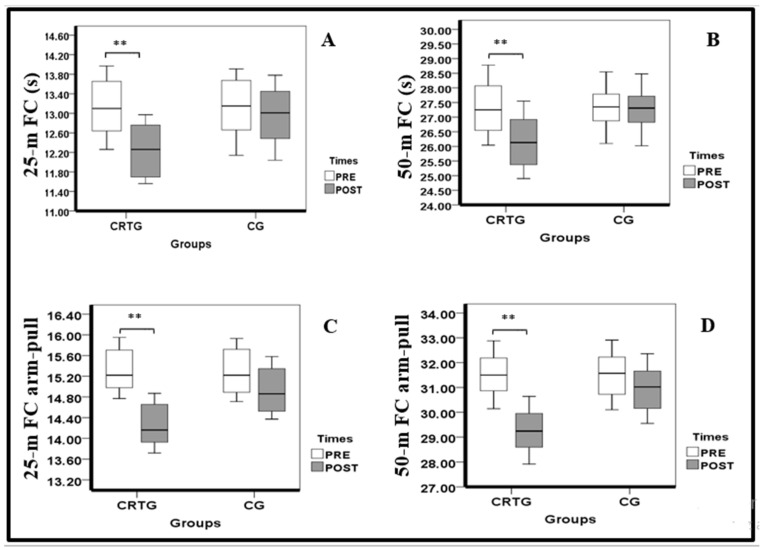
Box plots displaying the 25th and 75th percentiles, medians and whiskers extending from the minimal to maximal for the (**A**) 25-m FC, (**B**) 50-m FC, (**C**) 25-m FC arm pull and (**D**) 50-m FC arm pull in both groups (CRTG and CG) after 9 weeks of training (** *p* < 0.01). FC: front crawl.

**Figure 4 ijerph-18-10261-f004:**
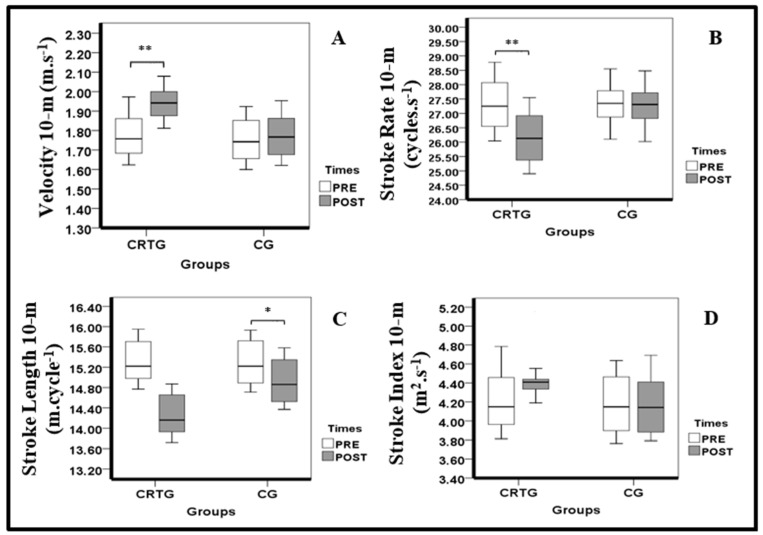
Box plots displaying the 25th and 75th percentiles, medians and whiskers extending from the minimal to maximal for the (**A**) velocity 10-m, (**B**) stroke rate 10-m, (**C**) stroke length 10-m and (**D**) stroke index 10-m in both groups (CRTG and CG) after 9 weeks of training (* *p* < 0.05; ** *p* < 0.01).

**Table 1 ijerph-18-10261-t001:** Participant’s characteristics (mean ± SD) from the CRTG: concurrent resistance training group and CG: control group.

Participant’s Characteristics	CRTG	CG
Age (years)	16.5 ± 0.30	16.1 ± 0.32
Height (cm)	174 ± 9.80	175 ± 9.70
Body mass (kg)	72.7 ± 5.30	73.6 ± 5.25
Competitive swimming experience (years)	6. 86 ± 0.33	6.78 ± 0.34
In-water resistance experience (years)	6.71 ± 0.30	6.60 ± 0.28
Dry land resistance (years)	4.74 ± 0.25	4.55 ± 0.27

**Table 2 ijerph-18-10261-t002:** Upper body dry land training program during the intervention and taper periods.

Periods	Week (Session)	Exercises	Sets × Repetition × Intensity; Recovery (R = 3-min)	Load Per Week (kg)
Intervention period	W1 (S1/S2)	BP	3 × 12 × 60% 1RM BP	1940.10 ± 97.47
		MBT	3 × 12 × 2 kg	144
	W2 (S3/S4)	BP	3 × 12 × 65% 1RM BP	2101.7 ± 105.60
		MBT	3 × 12 × 2 kg	144
	W3 (S5/S6)	BP	4 × 10 × 70% 1RM BP	2514.9 ± 126.35
		MBT	4 × 10 × 3 kg	240
	W4 (S7/S8)	BP	5 × 8 × 75% 1RM BP	2694.5 ± 135.38
		MBT	5 × 8 × 4 kg	320
	W5 (S9/S10)	BP	5 × 6 × 80% 1RM BP	2155.6 ± 108.30
		MBT	5 × 6 × 5 kg	300
	W6 (S11/S12)	BP	4 × 10 × 75% 1RM BP	2694.5 ± 135.78
		MBT	4 × 10 × 4 kg	320
Taper period	W7 (S13)	BP	5 × 6 × 80% 1RM BP	1077.8 ± 54.15
		MBT	5 × 6 × 5 kg	125
	W8 (S14)	BP	6 × 8 × 75% 1RM BP	1616.7 ± 81.23
		MBT	6 × 8 × 4 kg	192
	W9 (S15)	BP	5 × 10 × 70% 1RM BP	1571.8 ± 78.97
		MBT	5 × 10 × 3 kg	150

R: Recovery between sets and exercises. w: week, S: session, BP: bench press, MBT: medicine ball throw and 1RM: one-repetition maximum.

**Table 3 ijerph-18-10261-t003:** Quantification of the volume load in the intervention and taper periods.

Exercises	Total Load of IP (kg)	Total Load of TP (kg)	Decreased Load from IP to TP (%)
Bench press	14,101 ± 708.48	4266.4 ± 214.35	69.75%
Medicine ball throw	1468	467	68.19%

IP: intervention period and TP: taper period.

**Table 4 ijerph-18-10261-t004:** Changes in the maximum upper body strength, sprint swimming performance and kinematic variables between the pre- and post-tests.

Variables Measured	Groups	Pre-Test	Post-Test	Effect (95% CI)	Delta (%)	*p*	ES
1RM BP (kg)	CRTG	44.91 ± 2.23	50.36 ± 2.94	−5.46 (−7.79 to 3.12)	12.11 ± 1.79	<0.001	2.18 (large)
	CG	45.73 ± 2.37	46.82 ± 2.75	−1.09 (−3.37 to 1.19)	2.36 ± 1.10	0.331	0.44 (moderate)
25-m FC (s)	CRTG	13.14 ± 0.61	12.24 ± 0.55	0.90 (0.38–1.41)	−6.82 ± 0.66	0.002	1.62 (large)
	CG	13.13 ± 0.62	12.97 ± 0.59	0.16 (−0.38 to 0.70)	−1.21 ± 0.39	0.542	0.28 (moderate)
50-m FC (s)	CRTG	27.32 ± 0.93	26.17 ± 0.92	1.15 (0.33–1.98)	−4.22 ± 0.18	0.009	1.3 (large)
	CG	27.33 ± 0.76	27.26 ± 0.75	0.06 (−0.61 to 0.73)	−0.23 ± 0.06	0.845	0.09 (small)
25-m FC arm pull (s)	CRTG	15.33 ± 0.43	14.28 ± 0.42	1.05 (0.68–1.43)	−6.86 ± 0.16	<0.001	2.61 (large)
	CG	15.29 ± 0.46	14.93 ± 0.46	0.37 (−0.04 to 0.77)	−2.39 ± 0.09	0.077	0.83 (large)
50-m FC arm pull (s)	CRTG	31.52 ± 0.91	29.27 ± 0.91	2.25 (1.44–3.06)	−7.13 ± 0.23	<0.001	2.59 (large)
	CG	31.49 ± 0.99	30.93 ± 0.99	0.56 (−0.32 to 1.44)	−1.79 ± 0.09	0.196	0.60 (large)
V10-m (m.s^−1^)	CRTG	1.78 ± 0.12	1.94 ± 0.09	−0.16 (−0.26 to −0.07)	9.36 ± 2.55	0.001	1.67 (large)
	CG	1.76 ± 0.12	1.77 ± 0.11	−0.02 (−0.12 to 0.09)	0.90 ± 0.534	0.755	0.14 (small)
SR10-m (cycles.s^−1^)	CRTG	0.75 ± 0.04	0.85 ± 0.05	−0.10 (−0.14 to −0.06)	13.51 ± 4.22	<0.001	2.24 (large)
	CG	0.74 ± 0.04	0.75 ± 0.04	−0.02 (−0.05 to 0.02)	2.44 ± 0.472	0.317	0.46 (moderate)
SL10-m (m.cycle^−1)^	CRTG	2.36 ± 0.06	2.29 ± 0.01	0.07 (−0.01 to 0.14)	−2.86 ± 3.36	0.065	0.87 (large)
	CG	2.38 ± 0.03	2.35 ± 0.03	0.04 (0.01–0.06)	−1.50 ± 0.65	0.012	1.24 (large)
SI10-m (m2.s^−1^)	CRTG	4.22 ± 0.33	4.45 ± 0.26	−0.22 (−0.49 to 0.05)	5.52 ± 5.55	0.098	0.78 (large)
	CG	4.19 ± 0.33	4.16 ± 0.31	0.03 (−0.26 to 0.31)	−0.61 ± 1.10	0.847	0.09 (small)

V: velocity, SR: stroke rate, SL: stroke length, SI: stroke index, 1RM BP: one maximum repetition of the bench press, FC: front crawl, ES: Cohen’; s d (effect size) and *p*: *p*-value.

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
