# Peer review of "The Effect of Concurrent Resistance Training on Upper Body Strength, Sprint Swimming Performance and Kinematics in Competitive Adolescent Swimmers. A Randomized Controlled Trial"

_ijerph, 2021, doi:10.3390/ijerph181910261_

Round 1

Reviewer 1 Report

This manuscript is a study of the effect of concurrent resistance training on upper body strength, swimming performance, and kinematics in competitive swimmers. The article addresses an interesting topic; however, the manuscript needs some improvements before it can be published.

Title

The age of the participants of this study is adolescent. Please clearly indicate the participants of the study on the title. Ex) The effect of concurrent resistance training on upper body strength, swimming performance, and kinematics in competitive adolescent swimmers. A randomized controlled trial.

Introduction

Please add the content of the need for exercise training for athletes in their adolescence.

Materials and Methods

Please add a figure of the study design.

Please add a table about the physical characteristics of the subject.

Results

Please add a figure of variables with significant statistical differences.

Institutional Review Board Statement:

Please add the content.

Author Response

Author’s responses to the reviewers

Dear reviewers and editors,

We thank you for your valuable time and for the constructive and helpful comments. We carefully addressed all of your concerns and suggestions in the following point-by-point statement. Amendments to the manuscript were made whenever necessary.

Best Regards,

Dr. Yassine Negra (on behalf of all authors)

Comments and Suggestions for Authors

This manuscript is a study of the effect of concurrent resistance training on upper body strength, swimming performance, and kinematics in competitive swimmers. The article addresses an interesting topic; however, the manuscript needs some improvements before it can be published.

Reviewer 1 (changes were highlighted in YELLOW)

Comment 1: Title

The age of the participants of this study is adolescent. Please clearly indicate the participants of the study on the title. Ex) The effect of concurrent resistance training on upper body strength, swimming performance, and kinematics in competitive adolescent swimmers. A randomized controlled trial.

Authors’ reply to comment 2:

 Dear reviewer, we more clarified the title as suggested. Thank you.

Introduction

Comment 2:

Please add the content of the need for exercise training for athletes in their adolescence.

Authors’ reply to comment 2:

Dear reviewer, thank you for your constructive comments. We added sentences indicating the crucial importance of exercice training in youth swimmers (i.e., prevent injuries and enhance physical fitness). Thank you.

In addition, several scientific reviews and position articles [2, 3] revealed that resistance training represents a safe and feasible means in healthy adolescent to enhance muscular strength and motor skills and to prevent sport injuries. More specifically, it has been reported that well developed levels of muscle strength and power play an important role in achieving high swimming performance in adolescent and youth swimmers [2, 3].

Materials and Methods

Comment 2: Please add a figure of the study design.

Authors’ reply to comment 2: We added figure as suggested. Thank you

Comment 3: Please add a table about the physical characteristics of the subject.

Authors’ reply to comment 3: Added as suggested. Thank you.

Comment 4: Results

Please add a figure of variables with significant statistical differences.

Authors’ reply to comment 4: Added as suggested. Thank you.

Comment 5: Dear reviewer, done as suggested. Thank you

Institutional Review Board Statement: Please add the content.

Authors’ reply to comment 5:

Dear reviewer, the IRB statement was added in the first submission:

All procedures were approved by the unit of research (UR17JS01) of the higher institute of sports and physical education of Ksar Saïd, Tunisia, and were in accordance with the test version of the declaration of Helsinki.

Reviewer 2 Report

This study suggested interesting results and experimental suggestions. 
Also, this paper is well written with logical flow. 
However, I found some major limitations and suggest revision related these issues. 
This paper deserves to be published after major revisions.

page 3. 
A total of 22 subjects were randomly assigned to each of the 2 groups. Organize your age, height, weight, swimming history, etc. in a table and add a table so you can check it at a glance. Are there any differences between the two groups? If there is a difference between the two groups, it cannot be considered that the process was conducted under the same conditions.
Also, if the control group performed regular training and CRT program was additionally performed to the CRT group, of course, the CRT group performed additional training, so it is possible to make a difference in various results. 

Author Response

Author’s responses to the reviewers

Dear reviewers and editors,

We thank you for your valuable time and for the constructive and helpful comments. We carefully addressed all of your concerns and suggestions in the following point-by-point statement. Amendments to the manuscript were made whenever necessary.

Best Regards,

Dr. Yassine Negra (on behalf of all authors)

Reviewer 2 (changes were highlighted in Green)

Comment 1:

This study suggested interesting results and experimental suggestions. Also, this paper is well written with logical flow. However, I found some major limitations and suggest revision related these issues. This paper deserves to be published after major revisions.

Authors’ reply to comment 1: Dear reviewers thank you for your affirmative and constructive comments. The authors take and respond to your comment point by point. Thank you

Comment 2:

A total of 22 subjects were randomly assigned to each of the 2 groups. Organize your age, height, weight, swimming history, etc. in a table and add a table so you can check it at a glance.

Authors’ reply to comment 2:

Dear reviewer, we added subjects’ characteristics in a table as suggested. Thank you

Comment 3:

Are there any differences between the two groups? If there is a difference between the two groups, it cannot be considered that the process was conducted under the same conditions.

Authors’ reply to comment 3:

Dear reviewer, more details were added in the result section.

All participants received treatments as allocated. Table 4 displays mean and standard

deviation for all analyzed variables. There were no statistically significant differences in

baseline values between the 2 groups. In addition, no significant differences were found

between both groups with regard to chronological age, height, body mass.

We added also the statistical approach used to verify the baseline difference.

“Baseline between-group differences were computed through independent t-tests.

Comment 4:

Also, if the control group performed regular training and CRT program was additionally performed to the CRT group, of course, the CRT group performed additional training, so it is possible to make a difference in various results. 

Authors’ reply to comment 4:

“CRT Protocol was integrated into the training season for 9-week (6-week: intervention period, 3-week: taper period) in replacement of some swimming specific drills.”

Reviewer 3 Report

This study evaluated the effect of 9-week dry-land and standard training programs on strength and swim performance. Overall, the study design was sound and the statistical analyses were carried out appropriately. My comments are mostly minor, intended to improve the clarity of the study as it is presented.

Abstract: be sure to define all abbreviations before using them

Introduction: lines58-59 there seem to be a parentheses missing

Methods: the description of the training protocols is a bit confusing; perhaps a figure may be useful to provide a visual aid for the reader

Statistical analysis: please include which statistical software program was used

Section 3.3: please double-check for a few grammatical errors in this paragraph

Section 4.1: the authors neglect to bring up the principle of specificity here; it would be important to point out that the dry-land group did specific training to help improve the strength measures and this was not replicated in the control group

Author Response

Author’s responses to the reviewers

Dear reviewers and editors,

We thank you for your valuable time and for the constructive and helpful comments. We carefully addressed all of your concerns and suggestions in the following point-by-point statement. Amendments to the manuscript were made whenever necessary.

Best Regards,

Dr. Yassine Negra (on behalf of all authors)

Reviewer 3 (changes were highlighted in Grey)

Comment 1

This study evaluated the effect of 9-week dry-land and standard training programs on strength and swim performance. Overall, the study design was sound and the statistical analyses were carried out appropriately. My comments are mostly minor, intended to improve the clarity of the study as it is presented.

Authors’ reply to comment 1:

Dear reviewer, thank you for you affirmative comments and constructive suggestions.

Comment 2:

Abstract: be sure to define all abbreviations before using them.

Authors’ reply to comment 2:

Verified and modified as suggested. Thank you

Comment 3:

Introduction: lines 58-59 there seem to be a parentheses missing

Authors’ reply to comment 3:

Added as suggested. Thank you

Comment 4:

Methods: the description of the training protocols is a bit confusing; perhaps a figure may be useful to provide a visual aid for the reader

Authors’ reply to comment 4:

Dear reviewer, we added a figure to more explain and clarify protocols as suggested. Thank you..

Comment 5:

Statistical analysis: please include which statistical software program was used

Authors’ reply to comment 4:

Included as suggested. Thank you

Comment 5:

Section 3.3: please double-check for a few grammatical errors in this paragraph.

Authors’ reply to comment 4:

Dear reviewer, we double-check this paragraph and the whole article by a native speaker. Thank you.

Comment 6

Section 4.1: the authors neglect to bring up the principle of specificity here; it would be important to point out that the dry-land group did specific training to help improve the strength measures and this was not replicated in the control group

Authors’ reply to comment 6:

Dear reviewer, we clarified more this point as requested. Thank you

The results of this present study showed a large enhancement in the 1RM bench press test from pre-test to post-test (12.11 ± 1.79%) just after the CRT. However, no significant changes in muscle strength were detected after regular swimming training (i.e., CG).This corresponds with the principle of training specificity which dictates that training-related adaptations are larger when the training features (e.g., type of exercise, contraction mode, movement velocity) are aligned with the tested activity (David G Behm, 1995).

Reviewer 4 Report

Summary: 

This study had two groups of 11 high-school age swimmers do either standard swim training (control group) or swim training with combined resistance training (CRT group) for 9 weeks. Endpoints were measured at the start and end of the 9-week intervention. The CRT group in dryland exercises (bench press and medicine ball throws) as well as use hand paddles and a water parachute during a standardized sprint swimming set, whereas, the control performed no dryland training and did not use equipment during the standardized sprint swimming set. Endpoints were 1RM bench press, 25-m and 50-m swimming speeds, and kinematic variables in an arms-only trial. I commend the authors for undergoing a randomized controlled trial and a 9-week training intervention.

My main critiques of the study are as follows:

  1. By having the CRT group do both drylands and water-based resistance training it makes it hard to determine which exercise CAUSED the improved strength and performance. Based on this study design, we can’t discern which type of exercises were most beneficial to the swimmers. This could be mentioned in the limitations.
  2. The large amount of statistics reported and lack of figures detracts from communicating the main findings of this study. Likewise, much of the results states “significant main effects” without stating which group was higher. Consider removing some of the tables and adding graphs, specifically with regard to 1RM, swimming speeds, and pertinent measurements for the arms-only trials. In looking through your tables, it’s very hard to determine:
    1. Were any variables different at the beginning?
    2. Which became different by the end?
  3. You could probably simplify your method of data analysis. Did you consider analyzing your pre and post data separate? Or calculating %improvement and comparing the two groups by comparing %improvement with a t test? This would be a simpler method and I think would communicate your findings more directly. Showing a large improvement in one group and not in the other would convince most readers of the effectiveness of your intervention.
  4. Seeing the key endpoints on a figure (opposed to in tables) is a more effective way to communicate your findings and makes the results section easier to read.
  5. The methods would benefit from a diagram showing when the assessments were made and the 6-week training vs. 3-week taper. Or at least show the swim taper in a table as well.
  6. Were the two timekeepers blinded to which swimmers were in which groups? Which time was recorded? The average? This needs to be stated. Also, you state the stopwatch was started when the swimmer left the blocks (line 203). This is atypical. Shouldn’t the time be started when the timekeeper says “go.”
  7. Using a pull buoy won’t eliminate kicking. It will lessen the amount most swimmers kick but it will not eliminate without an ankle band. This should be clarified (line 207).
  8. Your endpoints were both sprint events, this is a limitation and will affect the conclusions from your study. You should emphasize that CRT improvement muscular strength and sprint swimming performance, but be careful about extrapolating to longer distance events. I suggest using “sprint swimming performance” as opposed to “swimming performance” throughout. This is also be included in the limitations paragraph (which should be expanded with explanations and not just a list – explain WHY are these limitations)

Author Response

Author’s responses to the reviewers

Dear reviewers and editors,

We thank you for your valuable time and for the constructive and helpful comments. We carefully addressed all of your concerns and suggestions in the following point-by-point statement. Amendments to the manuscript were made whenever necessary.

Best Regards,

Dr. Yassine Negra (on behalf of all authors)

Reviewer 4 (changes were highlighted in pink)

This study had two groups of 11 high-school age swimmers do either standard swim training (control group) or swim training with combined resistance training (CRT group) for 9 weeks. Endpoints were measured at the start and end of the 9-week intervention. The CRT group in dryland exercises (bench press and medicine ball throws) as well as use hand paddles and a water parachute during a standardized sprint swimming set, whereas, the control performed no dryland training and did not use equipment during the standardized sprint swimming set. Endpoints were 1RM bench press, 25-m and 50-m swimming speeds, and kinematic variables in an arms-only trial. I commend the authors for undergoing a randomized controlled trial and a 9-week training intervention.

My main critiques of the study are as follows:

 Comment:

  1. By having the CRT group do both drylands and water-based resistance training it makes it hard to determine which exercise CAUSED the improved strength and performance. Based on this study design, we can’t discern which type of exercises were most beneficial to the swimmers. This could be mentioned in the limitations.

Authors’ reply to comment :

Dear reviewer, we agree with your remark and we added the raised point as limitations of the study.

Comment:

  1. The large amount of statistics reported and lack of figures detracts from communicating the main findings of this study. Likewise, much of the results states “significant main effects” without stating which group was higher. Consider removing some of the tables and adding graphs, specifically with regard to 1RM, swimming speeds, and pertinent measurements for the arms-only trials. In looking through your tables, it’s very hard to determine:
    1. Were any variables different at the beginning?
    2. Which became different by the end?

Authors’ reply to comment:

Dear reviewer we tried to make the results clearer than in the first version. In addition we added more details about the baseline values of the anthropometric and physical fitness measures. We also add figures and table to make clearer the readability of the article.

All participants received treatments as allocated. Table 4 displays mean and standard deviation for all analyzed variables. There were no statistically significant differences in baseline values between the 2 groups. In addition, no significant differences were found between both groups with regard to chronological age, height, body mass.

  1. You could probably simplify your method of data analysis. Did you consider analyzing your pre and post data separate? Or calculating % improvement and comparing the two groups by comparing % improvement with a t test? This would be a simpler method and I think would communicate your findings more directly. Showing a large improvement in one group and not in the other would convince most readers of the effectiveness of your intervention.

Authors’ reply to comment:

Dear reviewer, thank you for your comment. However, by having 2 factors (i.e., group, and time). We think that our used statistical approach is more adequate than using a paired sample t-test or any other methods. In addition, we have changed the total structure of the results section to be simpler. Thank you

  1. Seeing the key endpoints on a figure (opposed to in tables) is a more effective way to communicate your findings and makes the results section easier to read.

Authors’ reply to comment: we have added a statistical figure and we have eliminated table 1 in section results.

  1. The methods would benefit from a diagram showing when the assessments were made and the 6-week training vs. 3-week taper. Or at least show the swim taper in a table as well.

Authors’ reply to comment:

 We have added a diagram of study design. The data of intervention and taper period are mentioned in table 2 and 3

  1. Were the two timekeepers blinded to which swimmers were in which groups? Which time was recorded? The average? This needs to be stated. Also, you state the stopwatch was started when the swimmer left the blocks (line 203). This is atypical. Shouldn’t the time be started when the timekeeper says “go.”

Authors’ reply to comment:

We clarified more these points as suggested. Thank you.

“The best performance established by the swimmers was taken into account in the statistical analysis”

The stopwatch was started when the swimmer left the blocks:

Authors’ reply to comment:

This method was referenced and used by Sammoud et al. 2021

  1. Using a pull buoy won’t eliminate kicking. It will lessen the amount most swimmers kick but it will not eliminate without an ankle band. This should be clarified (line 207).

Authors’ reply to comment:

Dear reviewer, we totally agree with you, but in this study we used pull buoys and we gave the order not to use leg movements. Therefore, the race and the leg movement were visually supervised. In addition, if any swimmer used his leg. The race was stopped and repeated after 5-m of a passive recovery.

  1. Your endpoints were both sprint events, this is a limitation and will affect the conclusions from your study. You should emphasize that CRT improvement muscular strength and sprint swimming performance, but be careful about extrapolating to longer distance events. I suggest using “sprint swimming performance” as opposed to “swimming performance” throughout. This is also be included in the limitations paragraph (which should be expanded with explanations and not just a list – explain WHY are these limitations)

Authors’ reply to comment:

We have changed “swimming performance” by “sprint swimming performance” in the all of the text. Thank you.

Round 2

Reviewer 2 Report

This study suggested interesting results and experimental suggestions. 
Also, this paper is well written with logical flow. 
However, I found some minor limitations and suggest revision related these issues. 
This paper deserves to be published after minor revisions.

Author Response

Author’s responses to the reviewers

The author’s replies to reviewer 2 were highlighted in green

Dear reviewer,

Thank you for your affirmative response.

However, the authors did not find any comments or suggestions concerning the next round. So the authors take the lead and try to double-check the article and found some grammatical errors that were changed and corrected. Thank you.